# SyrAgri: A Recommender System for Agriculture in Mali

**Jacqueline Konaté [1,*]**, **Amadou G. Diarra [1,2]**, **Seydina O. Diarra [1]** and **Aminata Diallo [1]**

[1]  Département d'Enseignement et de Recherche en Mathématique et Informatique, Faculté des Sciences et Techniques, Université des Sciences des Techniques et de Technologies de Bamako, Bamako BPE 3206, Mali; amadou.diarra@univ-grenoble-alpes.fr (A.G.D.); seydinaoumardiarra@gmail.com (S.O.D.); mimidiallo777@gmail.com (A.D.)

[2]  Laboratoire d'Informatique de Grenoble, Université de Grenoble Alpes, 38400 Saint-Martin-d'Hères, France

*  Correspondence: jacqueline.konate@usttb.edu.ml

**Abstract:** This paper focuses on recommender system for agriculture in Mali called SyrAgri. The goal is to guide and improve the quality-of-experience of farmers by offering them good farming practices according to their needs. Two types of recommendations are essentially taken into account: the recommendation of crops and the recommendation of farming practices based on some predefined criteria which are: yield, life cycle of the crop, type of soil, growing season, etc. SyrAgri also informs farmers about crop rotation and the similarity between different types of crops based on the following parameters: crop families, growing seasons and appropriate soil types. For the development of this system a hybrid recommendation approach was used: demographic, semantic and collaborative methods. Each method is adapted to a specific stage of a user's visit to the system. The demographic approach is first activated in order to offer recommendations to new users of the system, which resolves the concept of cold start (immediate inclusion of a new item or a new user in the system). The semantic approach is then activated to recommend to the user items (crops, agricultural practices) semantically close to those (s)he has appreciated. Finally, the collaborative approach is used to recommend items that similar users have liked.

**Keywords:** recommander systems; artificial intelligence; decision making; agriculture

## 1. Introduction

Agriculture is an important economic activity in developing countries. It represents about 30% to 60% of gross domestic product (GDP) and it employs 40% to 90% of the active population and is based on small family farms [1]. Agriculture also produces most of the basic foodstuffs and is a source of livelihood and income for Mali [2].

Farmers must be informed in real time on: rainfall, problems linked to global warming, the trend in seed varieties, the types of soil suitable for a given crop, crop rotations, agricultural best practices, etc. To do this, we opted for Artificial Intelligence approach which has renewed interest in recent years due to the significant increase of current computers capacities. Thus, prospects for the introduction of Artificial Intelligence in different sectors such as agriculture, health, finance, industry, etc.

Initially, Artificial Intelligence is seen as the science that aims to endow machines with a processing analysis capacity similar to that of humans [3]. In fact, Artificial Intelligence is now seen as a set of

technologies that provide benefits such as increasing the effectiveness and efficiency in performing certain tasks.

Artificial Intelligence applications in agriculture are often done using the Internet of Things (IoT) approach, which consists of providing agricultural machinery with information and communication technologies to monitor the environment. the operating condition of these machines and help in decision-making [4,5]. However, due to the poorly mechanized agriculture in developing countries, we did not selected this approach. Another approach to introducing Artificial Intelligence is to use recommendation systems to guide farmers to make decision in order to increase their yield. Indeed, Recommendation systems are an artificial intelligence system that is widely used in decision-making today. Indeed, it uses different techniques at the crossroads of various disciplines to establish models of analysis and data processing in order to facilitate decision-making using big data from various sources.

Currently, we always have to make choices before different situations. The size of these decision areas can be large. For example, YouTube records important number of videos daily. Statistics from unofficial sources estimate that 500 h of video are recorded on this platform every minute. Thus, we can deduce that 131 millions hours of videos are recorded per year [6]. In the absence of data from official sources, we will be content with these information which reveal, approximately, the immense size of the data that the Youtube platform contains. The list of possibilities available is therefore generally very important, evaluating these possibilities to find which is more suitable is a difficult task and can consume a lot of time. In order to overcome the problems of information overload and choice, recommender systems appeared in the early 1990s [7]. Recommender systems are similar to search engines (e.g., Google) but different from them in conceptual point of view. A search engine receives a request from the user, usually in the form of text and provides an ordered list of elements (web pages, images, videos, ...) in order to allow the user to quickly access content considered relevant among the large number of information available on the Internet. Unlike the search engine, a recommender system does not receive a direct request from the user, but offers them new possibilities by learning their preferences from these past behaviors [7]. A recommender system must therefore have access to a history of data which can be in several forms: notes, purchases, clicks on web pages, browsing histories, etc. From this information, the recommender system will be able to adapt the response to the user.

In order to contribute to the development of agriculture in Mali and to help the farmers, we decided to design a recommender system for decision-making in the field of agriculture. A recommender system is one capable to provide personalized recommendations or to allow the user to be guided to interesting or useful resources within a large data space [8]. Our contribution is twofold: firstly we designed a recommender system based on recommender engine and secondly we made some experiments to compare our system to agronomists advises.

The recommender engine consists of 4 modules: (1) Cosine Similarity Calculation module to compute the similarity between crops, (2) Aggregation module for aggregating values, (3) Fuzzy measure calculation module to determine user preferences and (4) Recommendations scheduling module to classify the cultivation practices.

In the remainder of this paper, we will first describe the problems we are faced, and review the literature to present our problem statement and research questions, followed by a background on the recommender systems. Then, we are going to present our research method and system design and implementation. Subsequently, we present our case study made with agricultural engineers and agricultural researchers. Finally, we conclude with the discussion outlining interpretations, contribution and limitations of our study and directions for further research.

## 2. Problem Statement and Research Questions

As explained in introduction, agriculture is the main component in malian economy. Thus, it's very important to develop this field by helping farmers to improve their yield. However, in Mali, the agriculture has lot of problems and they are many fold.

The majority of farmers has not educated in the agriculture domain. So they practice agriculture in traditional manner. This is not efficient and conducts to low yield. Furthermore, the choices of crops and soil types are difficult to do for farmers. They are face to many challenges such as: climatic hazards, degradation of natural resources, desertification, difficulty for accessing agronomists, lack of comprehensive information on crops, heavy agricultural manual operations, the difficulty of mobilizing and capitalizing on knowledge in the field [9].

To overcome these problems, our goal is to propose a recommender system helping to answer the following questions:

- How to guide the cultivators on the cultivation techniques?
- How to guide farmers on the choice of seed varieties?
- How to guide farmers to choose soil types, seasons for a crop?
- How to inform farmers about crop rotation?

## 3. Background

### 3.1. Recommender Systems

The ability of computers to make recommendations to users has been recognized early in the history of computer science [10]. A library system was a first step towards automatic recommender systems. This system was quite primitive and its use has remained very limited [10]. In 1992, Tapestry document recommender system appeared and it aims to recommend documents from newsgroups that might interest a group of users. The approach used was manual collaborative filtering. Then, GroupLens comes which uses automatic collaborative filtering [11]. In recent years, recommender systems became increasingly interesting in human–computer interaction, machine learning and information retrieval.

The purpose of recommender system is to provide relevant resources to user according to their preferences. So, the search time become reduced and the user also receives suggestions from the system to which they would not have spontaneously paid attention. Recommender systems have been defined in several ways. The most popular and general definition that we cite here is that from Robin Burke [8]: "A recommender system is a system capable to provide personalized recommendations or to allow the user to be guided to interesting or useful resources within a large data space." The two basic entities that appear in all recommender systems are User and Item. The "User" is the person who uses a recommender system, gives their opinion on various items and receives the new recommendations from the system. "tem" is the general term used to refer to what the system recommends to users. The input data for a recommender system depends on the type of filtering algorithm used. Generally, they are part of one of following categories:

- *Estimates*: also called notes, they express users' opinions on various items and are often represented by triplets (user; items; note). The set of triples (user; item; note) forms what is called the notes matrix. The couples (user; item) where the user did not give a score for the item are unknown values in the matrix. Table 1 shows an example of a rating matrix for 4 users and 4 films. Values marked as "?" indicate that the user has not given a notice [12].
- *Demographic data*: it refers to information such user's as age, sex, country, education, etc. [13]. This type of data is generally difficult to obtain and is normally collected explicitly.

- *Content data*: based on a textual analysis of the documents linked to the elements evaluated by the user. The characteristics extracted from this analysis are used as inputs to the filtering algorithm in order to deduce a user profile [14,15].

**Table 1.** Example of note matrix.

|          | Sanoudjè | Wari | Faro | TaaféFanga |
|----------|----------|------|------|------------|
| **Hamidou** | 4        | 3    | 2    | 4          |
| **Awa**     | ?        | 4    | 5    | 5          |
| **Oumar**   | 2        | 2    | 4    | ?          |
| **Seydou**  | 3        | ?    | 5    | 2          |

*3.2. Classification of Recommender Systems*

The most used classification is a classification according to two approaches: recommendations based on content and those based on collaborative filtering [16]. In addition to these two approaches, Robin Burke suggests considering three other approaches [8]: recommendation based on demographic data, recommendation based on knowledge (knowledge-based) and recommendation based on utility (utility-based). However, these three approaches are special cases of classical approaches. We present below the Content-based approaches then those based on the collaborative filtering and finally the hybrid approaches.

3.2.1. Content-Based Approaches

Content-based recommendations consist of analyzing the content of the items that are candidates for recommendation or the descriptions of these items [17]. Content-based recommendation methods use techniques largely inspired by the field of information retrieval. The main difference is the absence of explicit user requests. Content-based approaches rather infer user preferences and recommend items whose content is similar to the content of items they have liked before [17]. Thus, when new items are introduced into the system, they can be recommended directly, without requiring integration time.

The strengths of content-based approaches are user autonomy because they treat each user independently so that only the evaluations of the user himself are taken into account to build their user profile and make the recommendation [7]; immediate consideration of a new item because items newly introduced can be recommended before they receive an evaluation from a user [18].

Content-based recommendation approaches also have some drawbacks like: the limit of the content analysis because the accuracy of the recommendations is linked to the amount of information available to the system to discriminate between items appreciated and those not appreciated by the user [19]; over-specialization because the system can only recommend items that are similar to user profile and user can only receive recommendations close to items they have noted or observed in the past [20]; the user could start a problem because the integrating of new user is not immediate, they must evaluate some items before the system can interpret their preferences and provide them with relevant recommendations [21].

3.2.2. Approaches Based on Collaborative Filtering

Collaborative filtering is an approach based on sharing opinions between users. It uses the principle of "word of mouth", which has always been practiced by humans to build an opinion on a product or service that they do not know. The basic assumption of this method is that the opinions of other users can be used to provide a reasonable prediction of the preference of the active user on an item that they have not yet rated [7]. These methods assume that if users have the same preferences on one set of items, then they will likely have the same preferences on another set of items that they have not yet rated. Suppose,

for example, that Oumar's neighbors find the seed of "Gambiaka" rice profitable, they may find it worth trying. If, on the other hand, the majority of its neighbors consider it as failure, then it may be that they decide to refrain from using it. Collaborative filtering techniques therefore recommend, to the current user, the items appreciated by the users with whom they share the same tastes. We are talking about similar users. There are generally two main subfamilies of collaborative filtering: memory-based methods and model-based methods. Some examples of collaborative filtering approaches are Collaborative filtering based on memory [7], Collaborative filtering based on memory [22], Collaborative filtering based on a model [13], Clustering Model [23], K-Means [12], RecTree [24], Surprise effect (serendipity) [18].

The main strength of collaborative filtering approaches is that there is no limit of content analysis because they can process any type of item without any information on their content.

About inconvenient, collaborative filtering approaches do not provide user autonomy because the evaluations of other users can be taken into account to make the recommendation to another one [7]. In addition, collaborative approaches cannot recommend an item only if it has been previously assessed by a group of users [18].

### 3.2.3. Hybrid Approaches

A recommender system is hybrid when it combines two or more different recommendation approaches. The content-based recommendation and the collaborative recommendation have often been considered as complementary [25]. Content-based approaches have the advantage of being able to recommend new items not yet rated by a user, while collaborative filtering can only recommend an item if it has been rated by a number of users before. Content-based approaches require having item attributes in addition to an analysis step in order to extract and represent them, while collaborative filtering does not require access to item content in order to be able to recommendation. The hybridization of these two techniques in order to address the shortcomings of each technique used alone and take advantage of their strengths has been the subject of several research studies. The *'Fab'* system [25] is one of the first hybrid recommender systems. It combines collaborative filtering and a content-based approach to address both the problem of cold start for items and overspecialization. In this system, two criteria must be satisfied to recommend an item: its content must be similar to the user's profile, and it must be appreciated by the nearest neighbors. There are many ways to hybridize and no consensus has been reached by the research community. However, Burke has identified seven different ways of hybridizing [25]:

- **Weighted**: the score obtained by each of the two techniques is combined into a single result.
- **Switching**: the system switches between the two recommendation techniques depending on the situation.
- **Mixed**: the lists of recommendations from the two techniques are merged into a single list.
- **Feature combination**: the data from the two techniques are combined and transmitted to a single recommendation algorithm.
- **Feature augmentation**: the result of one technique is used as input to the other technique.
- **Cascade**: in this type of hybridization, a recommendation technique is used to produce a first classification of candidate items and then a second technique refines the list of recommendations.
- **Definition of a meta level**: this method is analogous to the method by increasing properties but it is the learned model which is used as input to the second technique and not the list result of the recommendations

## 4. Research Method

To address the questions outlined in the introduction section, we propose SyrAgri a web based recommender system, adaptable to user's profile and sensitive to its context. SyrAgri is a referral system aiming to help farmers for adopting good farming practices or receiving information on the types of crops they are interested in. Our research goal is twofold:

- Design of a recommender system for farmers
- Evaluate and compare it to expert recommendation results

### 4.1. SyrAgri Overview

SyrAgri system consisting can be used in following cases:

- First scenario: to recommend crops that may interest a user. This recommendation is made based on the location provided by user. If no crop in the system matches the user's location, the recommendation will be based on users who come from the same location as him/her.
- Second scenario: to recommend crops similar to what interests the user. For example, a user interested in growing millet may be recommended to grow sorghum since both crops are grown on the same type of soil (semi-sandy, sandy.). So (s)he can receive guides on sorghum practices.
- Third scenario: to get recommendations related to a chosen crop, to get evaluation based on preference criteria (yield, quality, life cycle of the crop), the growing location, the type of soil and the growing season.

The adopted architecture for SyrAgri system is Client-Server based. The core part of this architecture is our recommendation engine which contents four (4) modules as shown in Figure 1.

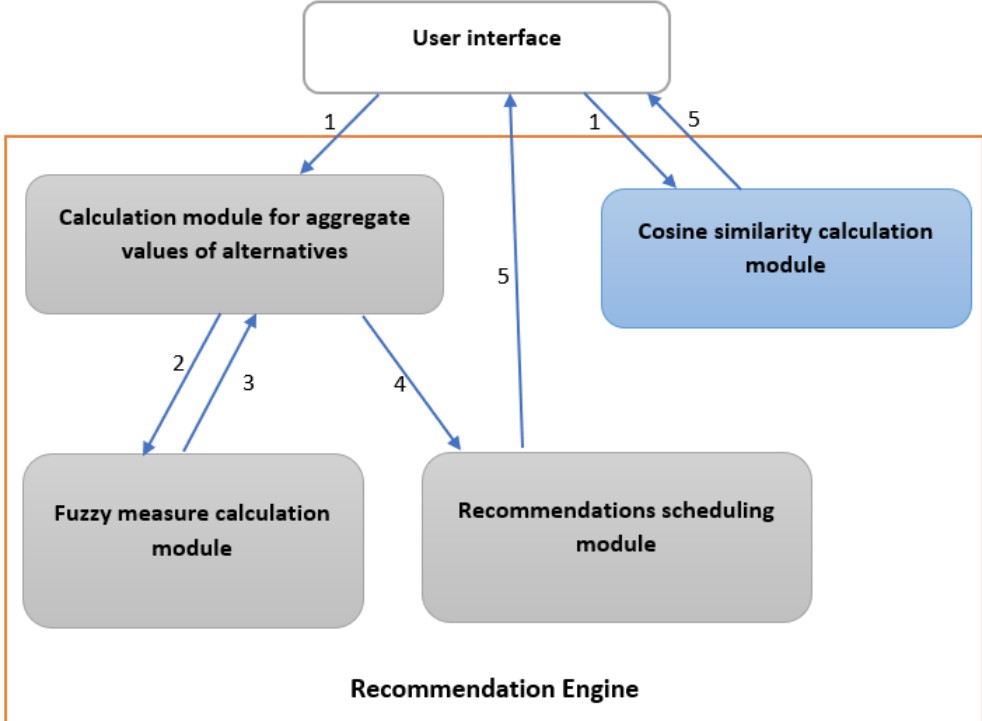

**Figure 1.** SyrAgri system architecture.

**Cosine Similarity Calculation Module**: this module is based based on Cosine similarity [21] and it allows to calculate the similarity (the degree of resemblance) between crops based on crop families, growing seasons and soil types suitable for cultivation.

**Module for calculating aggregate values of alternatives**: this module allows to calculate the integral of Choquet [26]. This involves calculating the overall score for each alternative (here the cultivation practices). To do this, the system receives as input parameter, the weights assigned by user to the preference criteria for the crop, the cultivation location, the cultivation season and the type of soil. The Algorithm 1 shows the execution steps of Choquet's integral.

**Fuzzy measure calculation module:** this module allows to determine a fuzzy measure called capacitance which corresponds to user preferences. The input parameters of this module are the weights that the user assigns to the preference criteria (Yield, Quality and duration of culture).

**Recommendations scheduling module:** this module allows to classify the cultivation practices that work when used by any user.

*4.2. Implementation*

To implement SyrAgri, we used Unified Modeling Language (UML) to model the concepts used in our knowledge base and their relationship. The details of this modeling are given in Appendix A.

4.2.1. Choquet's Integral

The integral of Choquet [26] was used for the recommendation of agricultural practices. It is a fairly flexible operator and allows to take into account the interactions between the criteria. The implementation of Choquet's integral involves determining a fuzzy measure called capacity that corresponds to user preferences.

To be able to use the integral of Choquet, we used the "Kappalab" package with the R language [27]. This package allows the manipulation of various types of functions such as the determination of fuzzy measurements.

- **Mathematical formula of Choquet's integral:**

   The integral of Choquet of $x = (x_1, \ldots, x_n) \in R^n$ compared to a capacity $\mu$ in N is defined by: $C_\mu$ **(x)** $= \sum_{i=1}^{n} x_{\sigma(i)} \lfloor \mu \left( A_{\sigma(i)} \right) - \mu \left( A_{\sigma(i+1)} \right) \rfloor$ ,

   where $\sigma$ is a permutation in N such as $x_{\sigma(1)} \leq \ldots \leq x_{\sigma(n)}$ ,

   $A_{\sigma(i)} = \{ \sigma(i), \ldots, \sigma(n) \}$ , for i $\in$ N, and $A_{\sigma(n+1)} = \varnothing$ .

- **Algorithm for integral of Choquet:**

   After calculating the fuzzy measure using the Kappalab package, we can calculate the overall score of each alternative (here the agricultural practices) in order to obtain a final ranking of alternatives. Algorithm 1 displays the algorithm used to implement the integral of Choquet.

---

**Algorithm 1:** Choquet's integral

---

**Data: double** alternatives[][], **double** criteria[], **double** mu[], **int** nbAlt, **int** nbC
**Result: double**[] aggregatedValues
**begin**
    **for** $i \in [1, nbAlt]$ **do**
        Swap the performance vector from smallest to greatest. Criteria should follow the same
         order $aggregatedValues[i] \longleftarrow 0$
        **for** $j \in [1, nbC]$ **do**
            $a_j \longleftarrow [j, ..., N]$
            $a_{j+} \longleftarrow [j+1, ..., N]$
            $aggregatedValues[i] \longleftarrow aggregatedValues[i] + alternatives[i][j] * (mu[a_j] - mu[a_{j+}])$
        **end**
    **end**
**end**

---

### 4.2.2. Similarity Cosine

Cosine similarity [21] is frequently used as a measure of similarity between two documents. Here, the documents represent descriptions of crops. This is to calculate the cosine of the angle between the vector representations of the elements or users to be compared. The similarity obtained, Simcosine, belongs to the interval [0,1]. In the case of two elements $i_1$ and $i_2$, the formula is:

$$Simcosinus(i_1, i_2) = \frac{vect(i_1) \cdot vect(i_2)}{\|vect(i_1)\| \cdot \|vect(i_2)\|}$$

## 5. Evaluation

To evaluate our system, we did two kinds of experiment: functional tests and qualitative tests.

### 5.1. Functional Tests

We are interested in functional tests to be able to test the different functionalities of our system. Functional tests make it possible to verify the compliance of the application developed with the initial specifications. Therefore, they are based on functional and technical specifications. To do this, we used the Selenium framework [28].

All functionalities of system have tested and validated after this step.

### 5.2. Qualitative Tests

Methodology

After testing the different functionalities of the SyrAgri system with Selenium software, we then made the qualitative evaluation of the SyrAgri system. This assessment was made using a method consisting of two main steps: collecting and analyzing data on cultivation techniques.

- **Data collection on cultivation techniques**

For the collection of data on cultivation techniques, we proceeded in two different ways:

    –    Collection of data through technical sheets on crops obtained from the malian Institute of Rural Economy (IER) and the Malian Ministry of Agriculture in order to populate the SyrAgri system database (see steps 1 and 2 of Figure 2).

    –    Data collection from experienced agronomists (agronomic data) through a form. It made it possible to make a comparative study of cultivation techniques in order to qualitatively assess the SyrAgri system (steps 3 and 4 of Figure 2).

- **Targeted agronomists**

For the filling of the forms we targeted ten (10) agronomists divided into two equitable groups: agronomists with more than 35 years of experience and agronomists with 3 to 5 years of experience.

This choice of two groups of agronomists is not fortuitous because it allowed us to obtain older data (technical and agricultural practices) from the first group and recent and ones from the second group.

- **Analysis and synthesis of the collected data**

After filling in the forms, we proceeded to the analysis and synthesis of the data from each group (step 5 of Figure 2). For this purpose, for each question, the answer which appears at least two times in the set of answers is selected.

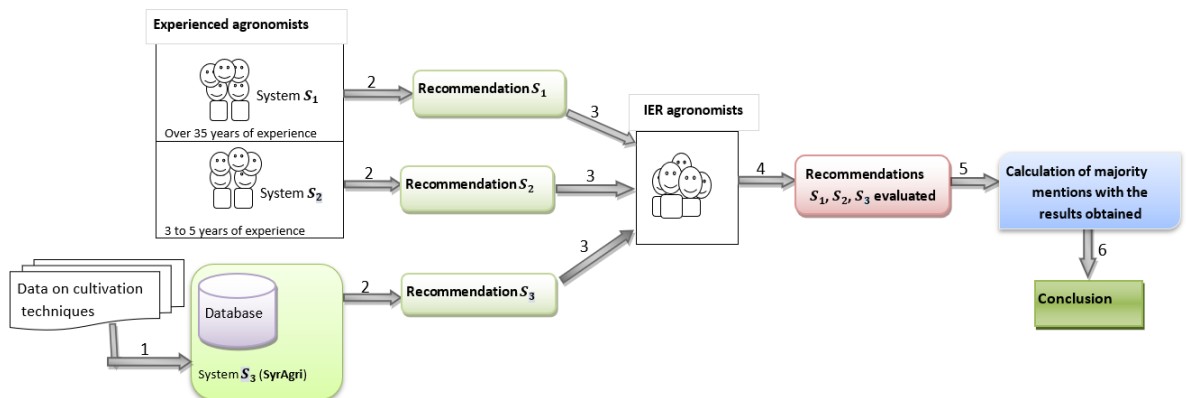

**Figure 2.** Illustrative diagram of the experimentation process.

The choice of at least two common answers out of all the answers is justified by the fact that there were questions of which only two answers were common out of the five answers received.

Following this analysis we obtained two systems: either the $S_1$ system (agronomists system with more than 35 years experience) and the $S_2$ system (agronomists system with 3-5 years experience) (step 2 of Figure 2).

Figure 2 presents the different stages concerning the qualitative evaluation of the SyrAgri system.

Below is the description of different steps:

1: Based on form provided in Appendix C, the two groups of engineering suggested commendations about the same list of cultures from technical sheets in step 1. The group of more experienced engineers is considered as system $S_1$ and the group of young engineers is the system $S_2$.

Technical sheets on crops from the Institute of Rural Economy and the Ministry of Agriculture have been used to collect data on cultivation techniques. The SyrAgri (system $S_3$) database has been populated by them.

2: the three system made recommandations about the cultures as Recommandation $S_1$, Recommandation $S_2$, Recommandation $S_3$.

3: The recommandations from step 2 are submitted to researchers from the Institute of Rural Economy in order to qualitatively evaluate them based on form shown in Appendix C.

4: Researchers evaluated the systems as recommandation $S_1$, recommandation $S_2$ and recommandation $S_3$ evaluated.

5: The majority mention method has been used to interpret the results of researchers evaluation.

6: Conclusion.

*5.3. Assessment Tools*

In order to interpret the results (step 5 in Figure 2), we used the majority judgment method [29,30].

Majority judgment is a method of voting for candidates (in this case systems), which consists in assigning each candidate endorsements in order to obtain the majority statement for each of them.

In our evaluation, the mention is used as metric and takes a value among (Excellent, Very-Good, Good, Fairly Good, Fair, Insufficient, To reject). The majority mention is the statement supported by a majority against any other mention [30].

So, by using this technique, the three systems are evaluated as follow:

- **Table of the mentions expressed on systems ($S_1$, $S_2$, $S_3$)**

In Table 2, the columns represent expressed mentions (Excellent, Very-Good, Good, Fairly Good, Fair, Insufficient, To reject) and on the lines we have the three systems (($S_1$, $S_2$, $S_3$).

**Table 2.** Table of expressed mentions for systems.

|  | Excellent | Very Well | Well | Pretty Good | Fair | Inadequate | To Reject |
|---|---|---|---|---|---|---|---|
| $S_1$ | 0 (0%) | 0 (0%) | 0 (0%) | 1 (20%) | 3 (60%) | 1 (20%) | 0 (0%) |
| $S_2$ | 0 (0%) | 0 (0%) | 1 (20%) | 3 (60%) | 1 (20%) | 0 (0%) | 0 (0%) |
| $S_3$ | 1 (20%) | 2 (40%) | 2 (40%) | 0 (0%) | 0 (0%) | 0 (0%) | 0 (0%) |

- **Graphs of the mentions expressed:**

Here we graphically represent the mentions expressed for each system. On the ordinate axis are represented the mentions and on the abscissa axis the number of votes allocated multiplied by 20% (scale considered).

In Figure 3, we note that the $S_1$ system obtained a single mention "Pretty Good" (20%), three mentions "Fair" (60%) and a single mention "Inadequate" (20%).

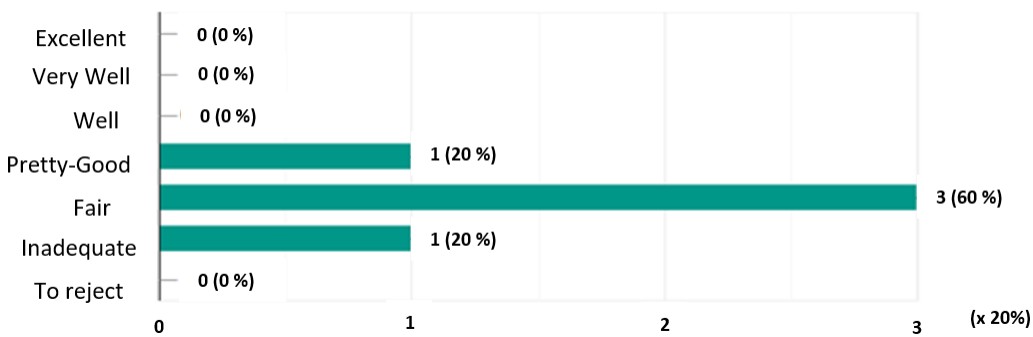

**Figure 3.** Mentions expressed for $S_1$ system.

Figure 4 shows that the $S_2$ system received a single "Well" rating of 20%, three "Fairly Good" ratings (60%) and a single "Fair" rating (20%).

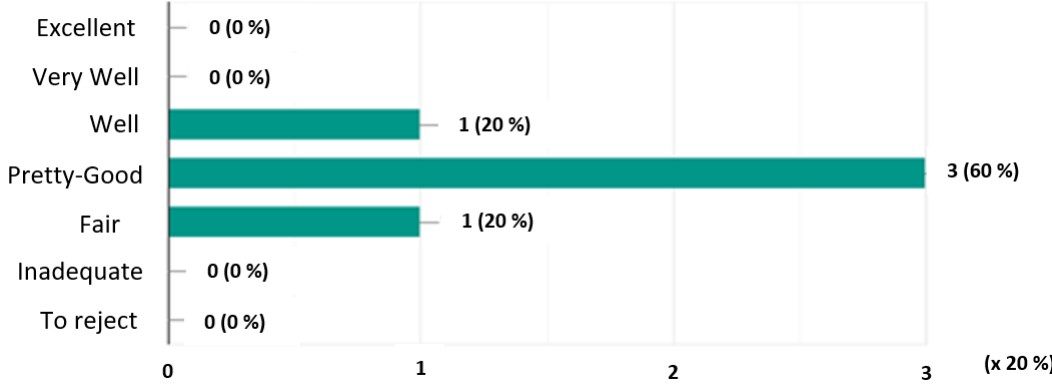

**Figure 4.** Mentions expressed for $S_2$ system.

In Figure 5, we see that the $S_3$ system has obtained a single mention "Excellent" or 20% , two mentions "Very good" (40%) and two mentions "Well" (40%).

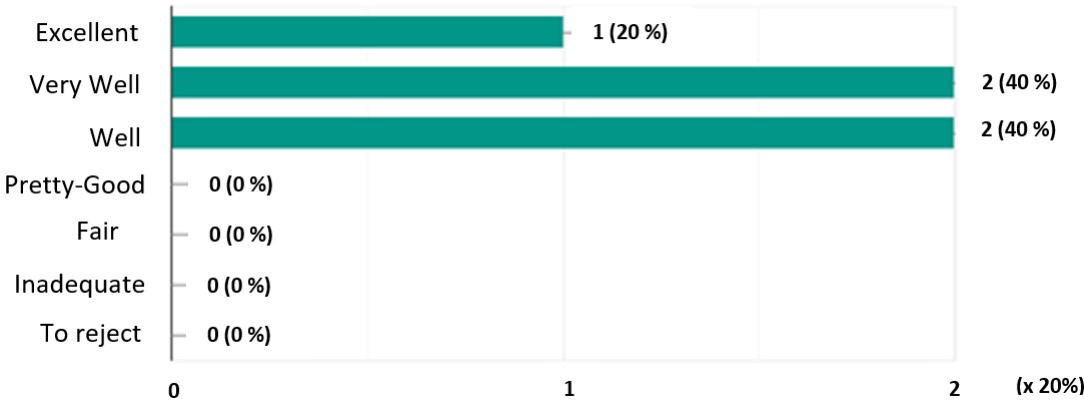

**Figure 5.** Mentions expressed for the $S_3$ system.

- **Calculating majority mentions**

  Considering Table 2, the mentions **'Excellent'**, **'Very-Well'**, **Good'**, **'Pretty-Good'**, **'Fair'** and **'To reject'** and let $M_j$ the majority mention of a system. The majority mention of a system in Table 2 is obtained as follows: (1) We check if the percentage value of the first column of the table is greater than or equal to 50% , then the mention of this column corresponds to the majority mention of the system otherwise we go to step 2; (2) We do the one-to-one sum of the values in percentages of the columns of the table from the left to the right (i.e., from the largest mention $E$ to the smallest 'R') until obtain at the level of a column a percentage greater than or equal to 50%. The mention of this column will then be considered as the majority mention of the system.

  Consequently, the majority of the $S_1$, $S_2$ and $S_3$ systems are as follows:

$M_j(S_1) = 0\% \text{ E} + 0\% \text{ T} + 0\% \text{ B} + 20\% \text{ A} + 60\% \text{ P} = 80\% \ (\geq 50\%).$

$M_j(S_2) = 0\% \text{ E} + 0\% \text{ T} + 20\% \text{ B} + 60\% \text{ A} = 80\% \ (\geq 50\%).$

$M_j(S_3) = 20\% \text{ E} + 40\% \text{ T} = 60\% \ (\geq 50\%).$

In conclusion, the majority mentions of the systems ($S_1, S_2, S_3$) are Fair, Pretty-Good and Very well respectively as indicated in Table 3.

**Table 3.** Table of majority mentions of systems.

|  | **Majority Mention ($M_j$)** |
| --- | --- |
| $S_1$ System | Fair |
| $S_2$ System | Pretty Good |
| $S_3$ System | Very Well |

- **Histogram with vertical rectangles of the mentions expressed for three systems**

In order to push further the qualitative comparison of the three systems, we deemed it necessary to graphically interpret the results (step 6 of Figure 2) of the three systems using a histogram with vertical rectangles. This graph allowed us to represent the mentions of the three systems in order to compare them. On the ordinate axis are shown the mentions expressed as a percentage and on the abscissa axis the different systems as shown in Figure 6. The mentions **excellent, very-well, well, pretty-good, fair, inadequate, to reject** are represented respectively by the bars in dark blue, green, blue, yellow, gray, orange, pale blue color.

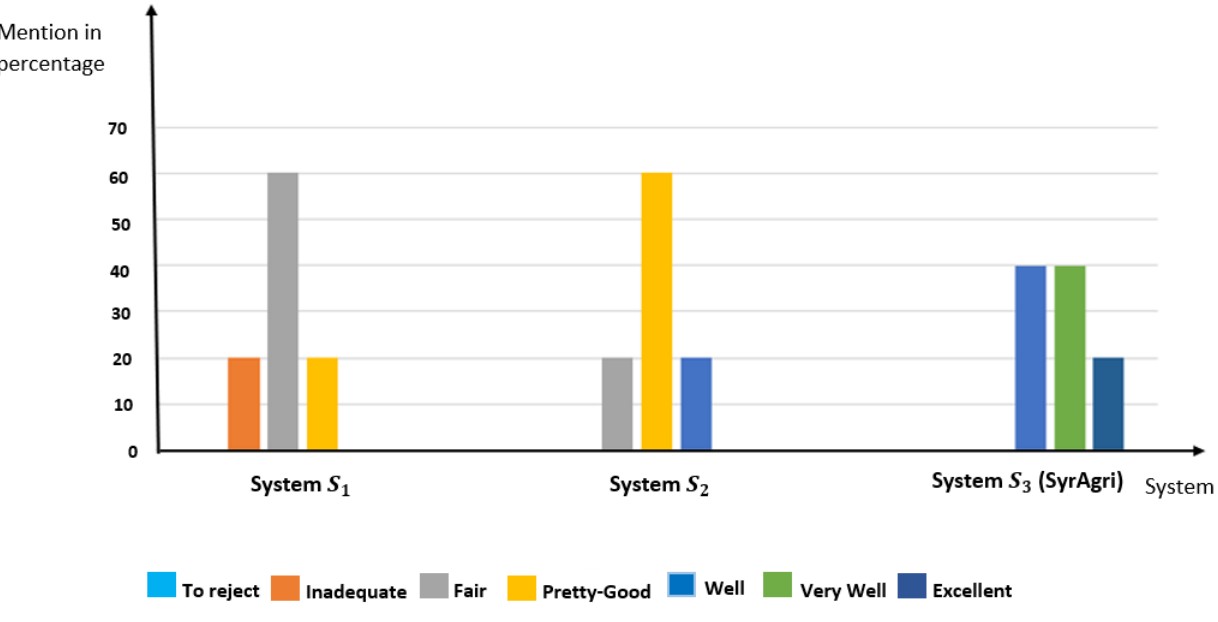

**Figure 6.** Graph comparing the mentions of three systems.

After analyzing this graph we notice that:

When we compare the mentions of the three systems, we find that the $S_3$ system was preferred to the other two systems ($S_1, S_2$) since it is the only one to have in its rating the three strongest mentions (excellent, very-good and good) among the existing mentions.

In conclusion, this process allowed us to qualitatively assess the SyrAgri system compared to the other two systems ($S_1$ and $S_2$). According to agricultural researchers from the Malian Institute of Rural Economy (IER: Institut d'Economie Rurale) and based on predefined criteria (yield, quality, life cycle of the crop, etc.). The SyrAgri system was preferred to the $S_1$ and $S_2$ systems. This is explained by the fact that its recommendations (seed varieties and most farming techniques) are precise and up to date, and in addition they are in perfect adequacy with the realities of climate change.

*5.4. Discussion*

### 5.4.1. Interpretation

This case study covers both a recommendation system and an assessment of the quality of recommendations in terms of good practice. In order to solve the challenges related to the lack of updated information on different crops, seasons, farming techniques, etc. We tested the system with 20 users who are divided into two categories of 10 users each and three categories of 5, 5 and 10 users respectively. These are 5 agricultural engineers with between 2 and 5 years of experience, 5 agricultural engineers with at least 35 years of experience and 10 researchers in agronomy. The 10 agricultural engineers divided into two groups of 5 provided data in response to the questions available in the question proposed in Appendix B. We considered the two sets of responses to be the recommendations of two systems $S_1$ and $S_2$. We submitted the same questions to our recommendation system and obtained responses constituting a third set of responses from an $S_3$ system. The three sets of recommendations were submitted to the group of 10 agronomists of a research institute for a qualitative assessment of the results from the three systems. According to the majority mention approach, system $S_3$ received the best evaluation (Appendix D).

### 5.4.2. Contribution

This study is the first of its kind to analyze the accuracy of the results and the usefulness of such a system in the field of agriculture in Mali. We have carried out a substantial bibliographic study on recommendation systems. This research effort has the merit of having succeeded in involving ten (10) agricultural engineers who know the field and ten (10) agricultural researchers with a number of years of experience which varies between 2 and 35 years. According to their assessment, our system offers relevant recommendations which cover an important spectrum of agricultural concerns: what is the suitable seed time for which type of soil? What are the ground preparation actions? What is the harvest period? What are similar cultures? Etc.

*5.5. Limitations and Perspectives*

We recognize, however, that our work has limits. First, only endpoint we have considered is the scientific accuracy of the recommendations. Secondly, the research was conducted in an intellectual specialist environment where people can read and write and are used to using the digital tool. When you go to a rural environment where the users are farmers who are not often literate and in particular are not driven by the digital wave, the results will be different. Thirdly, we only took demography into account because we were simply interested in the number of years of experience of our users. Finally, the study did not take into account the usability aspects of the system because we were much more concerned with the analysis of the quality of the recommendations rather than the properties of the human–machine interface. Future research will take into account the profiles of users (literate, non-literate), the adaptation of the system to rural populations, demographic aspects. Furthermore, we plan to develop a mobile application to better facilitate access to different recommendations for farmers. An improvement of the existing system is also envisaged as follows:

- Enrich the database with more technical data on crops: This will allow us to have several types of crops and a large number of recommendations on agricultural techniques.
- Develop new functionalities: recommendation on rainfall, generation of crop statistics in relation to regions of the country.
- Extend the system for livestock.

## 6. Conclusions

In this paper, we focused on recommender systems to support decision-making in the field of agriculture. Our goal was to design an expert system to improve the experience of farmers in their fields. We focused on recommender systems to help farmers choose the best practices for the success of their crops.

We have proposed a hybrid and context-sensitive recommendation approach that uses three recommendation methods: demographic, semantic and collaborative. The demographic approach is first used to offer recommendations to new users of the system, which resolves the concept of cold start (immediate inclusion of a new item in the system). The semantic approach is then activated to recommend to the user practices that are semantically close to those they have appreciated. The collaborative approach is ultimately used to recommend to the user practices that similar users have liked.

To carry out the application, we collected and analyzed the needs. Then we did the design and the implementation.

The objectives have been achieved. The current system offers features such as:

- Registration of a user;
- Authentication of a user;
- The registration of a crop;
- The registration of an agricultural practice;
- The suggestion of a crop;
- The recommendation of a crop similar to another;
- The recommendation of agricultural techniques for a given crop;
- The rating of a recommendation.
- Information on crop rotation.

**Author Contributions:** Conceptualization, A.G.D. and J.K.; methodology, J.K. and A.G.D.; software, S.O.D. and A.D.; validation, J.K. and A.G.D.; formal analysis, S.O.D. and A.D.; investigation, S.O.D. and A.D.; resources, A.G.D and J.K.; data curation, S.O.D. and A.D.; writing—original draft preparation, J.K. and A.G.D.; writing—review and editing, A.G.D. S.O.D. and A.D.; supervision, J.K. and A.G.D.; project administration, J.K. and A.G.D.; funding acquisition, J.K. All authors have read and agreed to the published version of the manuscript.

**Funding:** This research received no external funding.

**Acknowledgments:** We acknowledge the ministry of agriculture which provided the technical sheets for crops. We gratefully acknowledge agronomic engineers and researchers from malian institute for Rural Economy. A special thank towards Dabéré Jean Diassana who faciltated the contact with engineers and towards Anna Dembélé who did the same with researchers.

**Conflicts of Interest:** The authors declare no conflict of interest.

**Appendix A**

We present in this section the modeling details of SyrAgri. Figure A1 describes in Unified Modeling Language (UML) the concepts used in our knowledge base and their relationship.

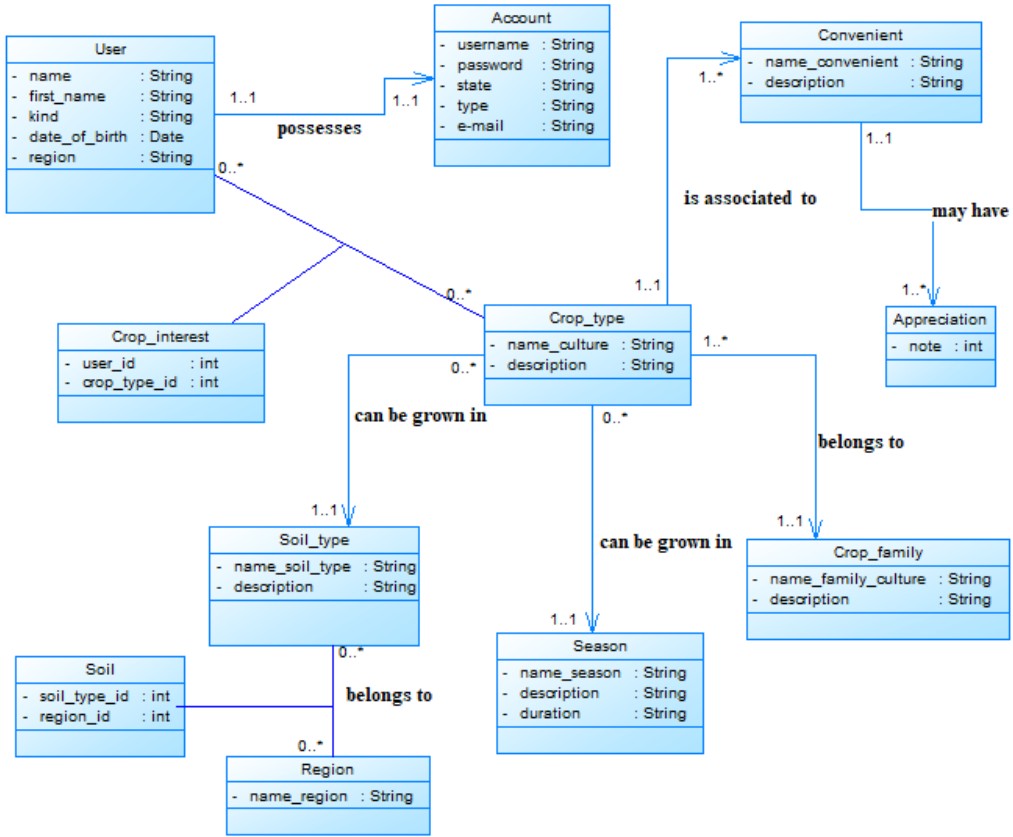

**Figure A1.** Class diagram.

A **user** can get one **account** and an **account** belongs to one **user**. A **user** can be interested in several **crop types** and a **crop types** can interest several users.

An **agricultural practice** is linked to a single **crop type** and a **crop type** is associated to several **agricultural practices**.

An **appreciation** concerns a single **convenient** and a **convenient** can have several **appreciations**.

A **crop type** belongs to a single **crop family** and a **crop family** has one or more **crop type**.

One **crop type** is associated to one **season**. Several **crop types** can be grown in the same **season**.

A **crop type** is associated to one **soil type**. Several **crop types** can be grown on the same **soil type**.

A **soil type** belongs to one or more **regions/locations** and a **region/location** can have several **soil types**.

**Appendix B**

In this section, we present our system through some of its interfaces. **Authentication interface** To access his/her personal space on the application, the user must first authenticate himself/herself through a form. This involves filling in the "Username" and "password"fields, provided during registration (Figure A2).

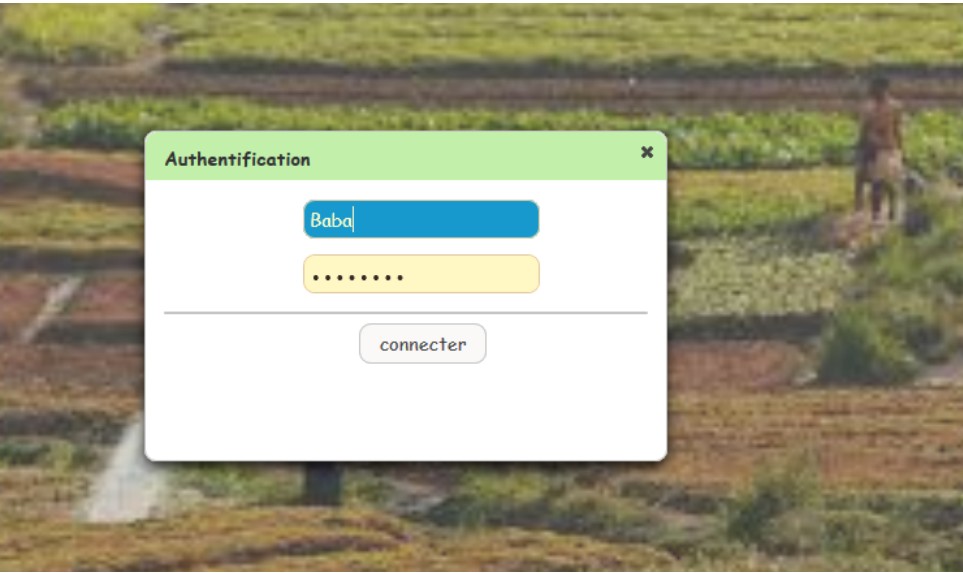

**Figure A2.** Authentication interface.

### Home Page Interface

For a new user, after authentication, the home page is displayed with the different menus and the crop recommendations (Figure A3).

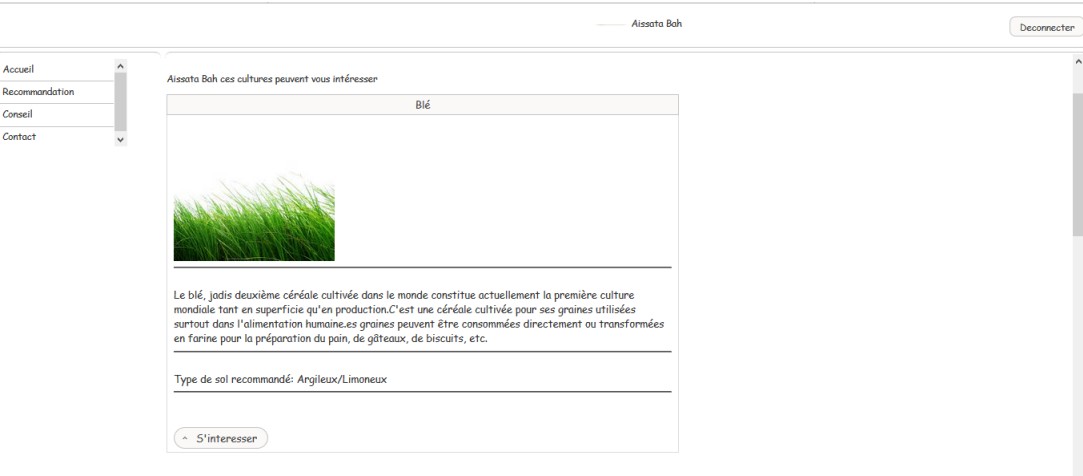

**Figure A3.** Interface-Home.

### Practices recommendation interfaces

The user can receive recommendations by clicking on recommendation in the menu. After choosing the crop, evaluating the criteria (yield, life cycle and quality of the crop) and providing information on its field, it will be displayed a list of recommendations for practice in relation to the chosen crop. So, they will be able to note their agricultural practices (Figures A4–A6).

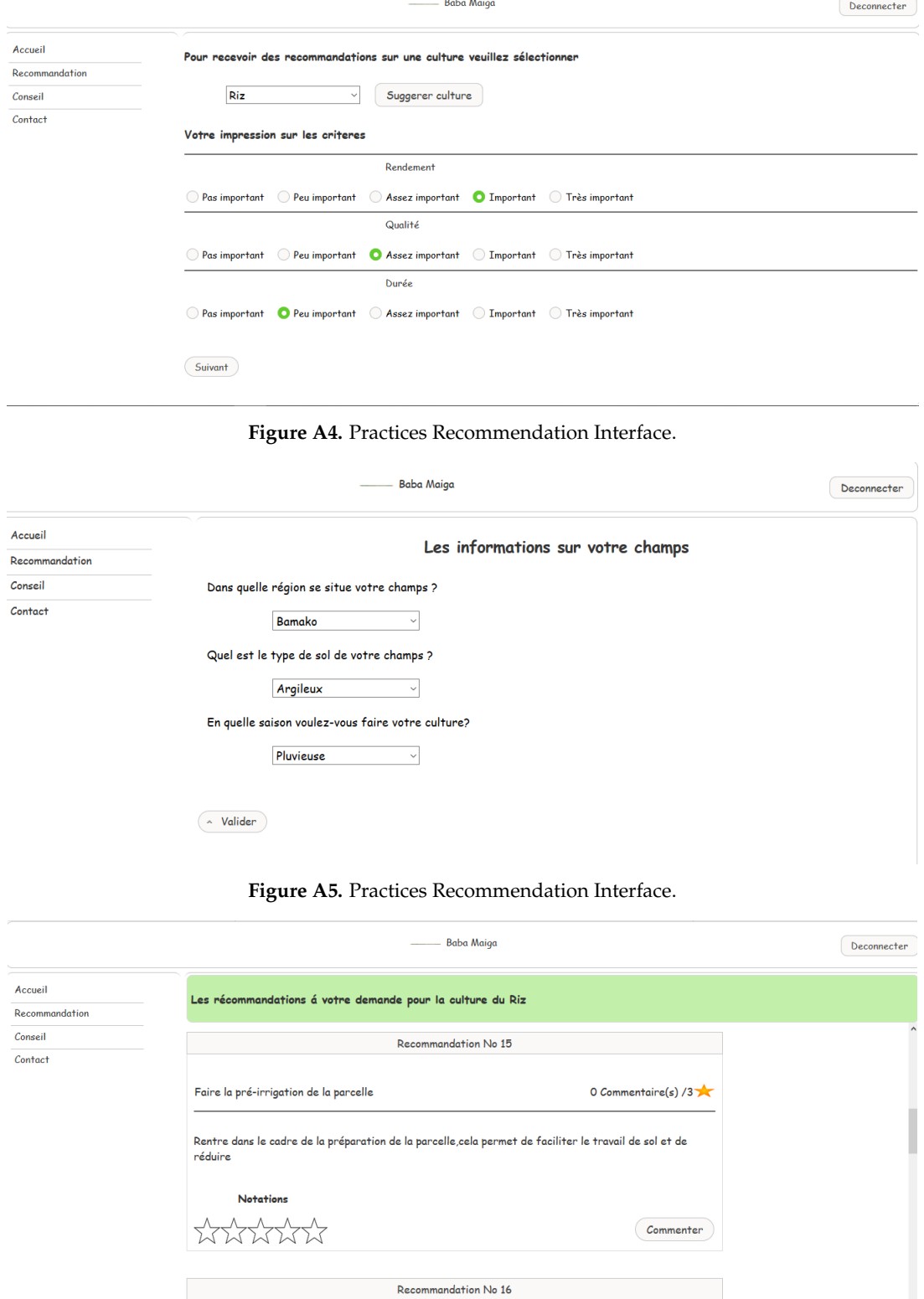

**Figure A4.** Practices Recommendation Interface.

**Figure A5.** Practices Recommendation Interface.

**Figure A6.** Practices Recommendation Interface.

**Crop registration**

The system administrator after having authenticated, can register a crop in the database by filling in the add form (Figure A7).

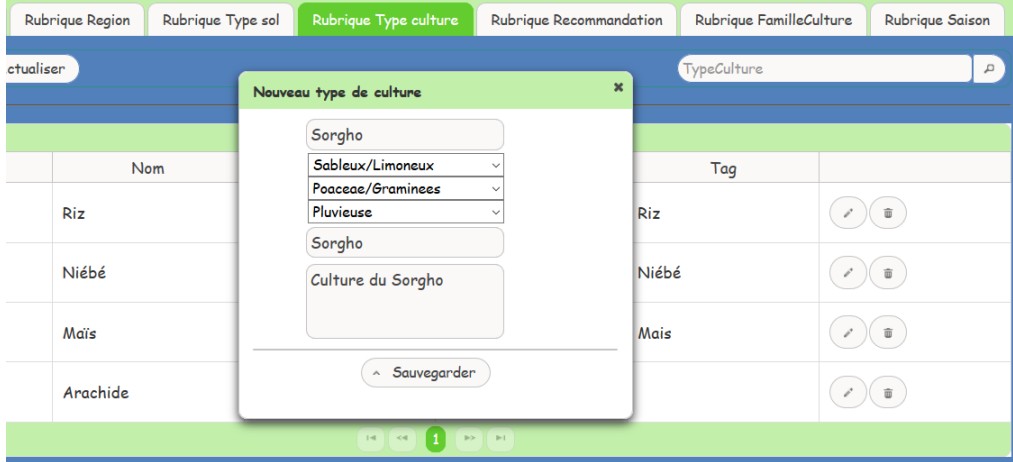

**Figure A7.** Crop registration interface.

**Appendix C**

In this section, we present the form for collecting data during the testing phase made by agronomic engineers.

# SyrAgri system evaluation sheet

Last name:

First name:

Number of years of experience:

Home institution:

### Description of SyrAgri system:

SyrAgri is a recommendation aid system for decision-making in the field of agriculture designed and developed by master computer science students of the Faculty of Sciences and Techniques of the University of Sciences, Techniques and Technologies of Bamako (FST-USTTB). Its objective is to improve the experience of farmers by offering them good farming practices according to their needs. The system retrieves input data such as the preference criteria for the crop (yield, quality and duration of the crop), the crop region, the crop season and the type of soil used for the crop, and then offers user output data.

If the system does not have specific information on the type of crop supplied by the user, it is based on the similarities between this type of crop and others, taking as parameters the crop families, the seasons of crop and soil types suitable for cultivation.

We are submitting this form to you in order to gather information on some crops. Your answers will then be compared to those of the SyrAgri system for a qualitative evaluation.

For your recommendations, please, consider that interests are best yield, then best quality and finally a short cultivation time.

**I. Your recommendations for the success of the following crop: Rice**

a.  Which geographic regions do you recommend for this crop?

- Region 1:
- Region 2:
- Region 3:
- Etc:

b.  What are the growing seasons?

- Season 1:
- Season 2:
- Season 3:
- Etc:

c.  What types of soil do you reccommend?

- Type 1:
- Type 2:
- Type 3:
- Etc:

d.  What types of seeds do you reccommend?

- Type 1:
- Type 2:
- Type 3:
- Etc:

e.  What are the site preparation actions that you recommend?

- Action 1:
- Action 2:
- Action 3:
- Etc:

f.  What measures do you recommend for soil fertilization?

- Measure 1:
- Measure 2:
- Measure 3:
- Etc:

g.  When and how do we sow?

- Period:
- Action:

h.  What are the appropriate maintenance actions for the following periods?

| Period | Action |
|--------|--------|
|        |        |
|        |        |
|        |        |
|        |        |

i.   When should we harvest?

j.   How should we harvest?

- Action 1:
- Action 2:
- Action 3:
- Etc:

## II. From your experience, what are the similar of the following crops?

- Rice:
- Corn
- Cowpea:
- Okra:
- Sorghum:

## Appendix D

In this section, we present the evaluation form used by agronomic researchers.

## Assessment form for recommendation systems ($S_1, S_2, S_3$) for agriculture

Last name:

First name:

Number of years of experience:

Home institution:

## Description:

The systems $S_1$, $S_2$, $S_3$ are recommendation support systems for decision-making in the field of agriculture. Their objectives are to improve the experience of farmers by offering them good farming practices according to their needs. The systems retrieve input data such as the preference criteria for the crop (yield, quality and duration of the crop), the crop region, the crop season and the type of soil used for the crop, and then propose user output data.

In case the systems do not have specific information on the type of crop gave by the user, it is based on the similarities between this type of crop and others by taking as parameters the crop families, the seasons of crop and soil types suitable for cultivation.

We are submitting this form to you to collect your impressions of the recommendations made by these systems. Your answers will then be used to make a qualitative comparison between these systems.

**I. Comparison between data from system $S_1$ and those from system $S_3$**

In the Interpretations column, please provide information on the recommendations for the proper functioning of the following crop: Rice.

**Table A1.** Example of note matrix.

| | System S1 | System S2 | System S3 | Interpretations |
|---|---|---|---|---|
| **Growing region** | Ségou | Ségou | Ségou | |
| **Growing season** | Rainy saison | Rainy saison | Rainy saison | |
| **Type of soil** | Clay soil | Clay soil | Clay soil | |
| **Seed varieties** | Gambiaka, Neureka, ADNY. | ADNY, Kogoni, Neureka, Gambiaka. | ADNY-11 (6 t/ha, 120 days), Gambiaka (5 t/ha, 160 days), Telimani (7 t/ha 115 to 125 days), Neureka, Kogoni-91-1. | |
| **Ground preparation actions** | Complete plowing of the field. Harrow of the field, Putting in mud and finally leveling. | Clean the plot if necessary; Plow to a depth of 10–15 cm after irrigation; Harrow / spray then mud. | Complete plowing the land (10 to 15 cm deep), Leveling, Establishment of a nursery (plowing, leveling, maintenance) An adequate irrigation and drainage network to ensure water control. | |
| **Soil fertilization actions** | Add organic manure (cattle manure, waste) before plowing; Harrow of the field, Putting in mud and finally leveling. | Addition of 20–30 t/ha of organic manure well decomposed before plowing; Addition of 100 kg/ha of DAP or 100 kg/ha of cereal complex; Add 200 kg/ha of urea divided into two applications (15 days apart), the first of which at the start of tillering. | Add 250 kg/ha of ground manure at the time of plowing. Add 100 kg/ha of DAP immediately after sowing and 200 kg/ha of emergent urea. | |
| **Sowing actions** | Establishment of a nursery: The nursery sowing is done on the fly and to cover with a layer of 0.5 cm of straw to conserve the humidity of the environment. Harrow of the field, Transplanting: it is done in line with a transplanting density of 0.20 m × 0.20 m and the number of plant / poquet is from 2 to 3 plants without prickling. | Beginning of June or Mid- June: it can be done indirectly (the nursery) on the fly with a density between 30-80 kg of seeds per ha. End of June: Transplanting is done 20 to 25 days after the online nursery due to 1 to 2 plants per poquet. The distance between the lines varies between 20–40 cm. | Nursery location: Choose a sunny location close to the bedding plot and the water point. Nursery preparation: Plowing (7 ares per ha to transplant) followed by harrowing then the making of 10 m × 1 m boards independent of each other. Seeds: 60 kg to be pregerminated +120 g of fungicide before sowing in the nursery. Nursery seedlings: On the fly and cover with a layer of 0.5 cm of straw to conserve the humidity of the environment. | |

**Table A1.** *Cont.*

|  | System S1 | System S2 | System S3 | Interpretations |
|---|---|---|---|---|
| **Sowing actions** |  |  | Nursery manure: 1 to 2 wheelbarrows of farmyard manure for 10 m$^2$ and 250 g for 10 m$^2$ of super triple or potassium chloride on the fly, preferably as background manure.<br><br>Number of plants per poquet:2 3 to plants without prickling.<br><br>Transplant density: 0.20 m × 0.20 m. |  |
| **Field maintenance actions** | Weeding: Do weeding (manual, chemical mechanical) or herbicide fifteen days after sowing.<br><br>250 g for 10 m$^2$ of super triple or potassium chloride on the fly, preferably as background manure.<br><br>Irrigation: On the 15th day of weeding, put in water (maintaining the water slide at 5–10 cm cube depending on the height of the plants). | 2-3 days after transplanting, relining preherbicide.<br><br>20–25 days after transplanting: Dismantling of the plants, transplanting 1 to 2 plants per pocket (if necessary)<br><br>15 days after transplanting, do the 1st weeding and the first slice of urea plus the whole DAP<br><br>At the time of the initiation of the panicles, make the contribution of the 2nd slice of urea.<br><br>In case of attack of diseases or insects phytosanitary treatment. | Maintenance: 2 weeding including the second 15 days after the first (manual, chemical mechanics), and phytosanitary treatments on request.<br><br>Irrigation: 20,000 m$^3$ on average with a blade of water equal to a third of the size of the plant with each supply. Empty 2 to 3 days to the 15th day for weeding, fertilizing and aerating the soil. |  |
| **Harvest periods** | When the ears begin to have a golden color and become dry and hard to the touch. | Harvest at 2/3 of maturity, that is to say when the seeds become hard and are no longer milky. | Harvesting should take place at a time when the grains have a suitable moisture level so that they do not break. This corresponds to a period of 30 to 40 days after heading, depending on the variety. |  |
| **Harvest actions** | Harvesting is done by fossil or using the harvester. | Harvesting can be manual with a sickle or mechanical with the combine harvester, binder harvester.<br><br>The sheaves are dried 3 to 4 days on the ground before hilling. | Harvesting can be done mechanically with a combine harvester, a baler harvester or manually with a sickle 30 to 50 cm below the panicle so as not to get tired. The sheaves are dried 3 to 4 days on the ground before threshing. |  |
| **Similar crops** | Wheat, Corn. | Sugar cane, Wheat, Corn. | Wheat, Corn, Sorghum, Mil. |  |

**II. Scoring of the three systems ($S_1$, $S_2$ and $S_3$)**

After reading the various recommendations made by the three systems, please assign a rating to each of them.

**NB:** Put a cross in the corresponding boxes.

| Mentions ⟍ System | to Reject | Inadequate | Fair | Pretty-Good | Well | Very-Well | Excellent |
|---|---|---|---|---|---|---|---|
| System S1 | | | | | | | |
| System S2 | | | | | | | |
| System S3 | | | | | | | |

<u>Signature</u>

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
