# Peer review of "SyrAgri: A Recommender System for Agriculture in Mali"

_information, doi:10.3390/info11120561_

Round 1
Reviewer 1 Report
# The highlights of the article should be provided. Each highlight should be given in a short sentence.
# In state of the art, the authors need to discuss the works instead of only mentioning that author `A' did this and author `B' did this. So in discussion of the related works the authors should consider this.
# As mentioned in the paper "Finally, the collaborative approach is used to recommend items that similar users have liked. " that sounds like keyword extraction. it should be highlighted in your work, and also the article can be enriched using works like " Main large data set features detection by a linear predictor model "(2014).
# The article needs English language checking. It is suggested to be reviewed by an expert of the English language. # The quality of images should be improved. # check he caption of Figure 11. what is "og"? # It is suggested a review of applications of Information Technology in Agriculture to be added. In this regards, the following works which are about application of artificial intelligence in fault diagnosis of agriculture machinery are very useful and suggested to be cited: “Economic data analytic AI technique on IoT edge devices for health monitoring of agriculture machines (2020)”, “Economic IoT strategy: the future technology for health monitoring and diagnostic of agriculture vehicles (2020)”Author Response
Responses to Reviewer 1 Comments
Point 1: The highlights of the article should be provided. Each highlight should be given in a short sentence
Response 1: We highlighted the keys ideas in the new version of our paper (see introduction section)
Point 2: In state of the art, the authors need to discuss the works instead of only mentioning that author `A' did this and author `B' did this. So, in discussion of the related works the authors should consider this.
Response 2: The state of the art has been modified (see background section).
Point 3: As mentioned in the paper "Finally, the collaborative approach is used to recommend items that similar users have liked. " that sounds like keyword extraction. it should be highlighted in your work, and also the article can be enriched using works like " Main large data set features detection by a linear predictor model "(2014).
Response 3: We focused on related work about collaborative filtering reason that why we didn’t mention work on features detection, keyword extraction. We think collaborative filtering already address these problems.
Point 4: The article needs English language checking. It is suggested to be reviewed by an expert of the English language.
Response 4: English language checking has been done in the new version.
Point 5: The quality of images should be improved. Check he caption of Figure 11. what is "og"?
Response 5: These problems have been fixed in new version of paper.
Point 6: It is suggested a review of applications of Information Technology in Agriculture to be added. In this regard, the following works which are about application of artificial intelligence in fault diagnosis of agriculture machinery are very useful and suggested to be cited: “Economic data analytic AI technique on IoT edge devices for health monitoring of agriculture machines (2020)”, “Economic IoT strategy: the future technology for health monitoring and diagnostic of agriculture vehicles (2020)”
Response 6: We already took into account this point. However, our work addresses some problems related to agriculture in Mali like yield optimisation, lack of information to farmers, by using recommendation system. Also, by considering the developing countries agriculture conditions, where the agriculture is not sufficiently mechanised and IoT technologies are not in point, we used systems recommendations approaches which are easy to adapt to our context (see lines 33 – 42).

Reviewer 2 Report
As claimed, this article proposes a hybrid recommender system named SyrAgri to suggest the farmers with good farming practices based on their needs. The system takes advantage of demographic, semantic, and collaborative recommendation modules and adjusts certain parameters including Crop families, growing seasons, and appropriate soil types.
The authors must improve the readability of the paper and remove extra spaces, spelling, and grammatical errors. Better word choices are also advised to improve the readability of the paper.
Example: Currently, we always have to make choices. Which activity to do before in day ? Which company to travel for ? Which seed to sow? Etc.
Another example, line 19: videos dayly --> a very basic spelling error --> correct: Daily
The paper should be written in a more concise way. For example, I refer to the first paragraph in the introduction section is which quite general and doesn’t have any relevance to SyrAgri.
The significant novelties of the proposed system are not evident. What can be explained more about the contribution of the algorithm in the introduction section?
Related work should include the effect of the recommendation parameters, including time and location. For example, the time of farming can differ based on the seasons.
Example paper on this regard:
Leveraging multi-aspect time-related influence in location recommendation
or similar concept in neural networks: SoulMate: Short-text author linking through Multi-aspect temporal-textual embedding.
Related work is over-explained. At some points, one paragraph explains a single paper. Section 2.4. Some examples of recommender systems is not really necessary in the related work.
In section 3.SyRAgri recommender system, the authors explain the application side of the work. While in research we need to start the research problem
Figure 1 is very naïve from a research perspective.
Use case diagrams and the database design details are related to the application, not the research. Which research problem we address in a paper is important.
The scientific novelty of the paper is missing. The Cosine similarity and Choquet’s integral are not included as a contribution to this paper. So no need to over-explain them.
In a research paper, we include the algorithms, not the codes: Example of the wrong usage: Algorithm for integral of Choquet
A single citation is adequate for Cosine similarity, therefore section 3.2.3 is not required to be in the paper.
In a research paper, we do not provide a demo of the application. Figure 6. Test of adding crops
You need to add a problem statement section to the paper which also includes the formal mathematical definitions.
Figure 7 must include information about the scientific modules, not schematic components.
The evaluation section must include the evaluation metrics, the evaluation method, and the baselines.
In general, it is not clear what novelty has been applied in this paper to promote recommendation systems from a scientific perspective.
The authors should do an experiment to compare the performance of the proposed method with another state of the art or at least common methods.
Appendix of an article usually includes the mathematical proofs.
The scientific contribution of this paper is very limited.
Author Response
Response to Reviewer 2 Comments
Point 1: The authors must improve the readability of the paper and remove extra spaces, spelling, and grammatical errors. Better word choices are also advised to improve the readability of the paper.
Response 1: These problems are fixed in the new version of our paper. We removed spaces in annexes.
Point 2: The paper should be written in a more concise way. For example, I refer to the first paragraph in the introduction section is which quite general and doesn’t have any relevance to SyrAgri.
Response 2: We highlighted SyrAgri in the new version. The introduction has been reworked from scratch and new things have been added to better focus the section on the topic covered in the article.
Point 3: The significant novelties of the proposed system are not evident. What can be explained more about the contribution of the algorithm in the introduction section?
Response 3: We have modified the presentation of SyrAgri system by highlighting our contribution (see sections 3 and 4).
Point 4: Related work should include the effect of the recommendation parameters, including time and location. For example, the time of farming can differ based on the seasons. Example paper on this regard: Leveraging multi-aspect time-related influence in location recommendation or similar concept in neural networks: SoulMate: Short-text author linking through Multi-aspect temporal-textual embedding.
Response 4: We already took into account recommendation parameters such as geographic location (region), soil type, crop family, and season. They are highlighted in the new version. The suggested paper has been introduced as reference 15.
Point 5: Related work is over-explained. At some points, one paragraph explains a single paper. Section 2.4. Some examples of recommender systems is not really necessary in the related work.
Response 5: State-of-the-art reduction efforts have been made (see section 2 which is no longer two pages).
Point 6: In section 3.SyRAgri recommender system, the authors explain the application side of the work. While in research we need to start the research problem. Figure 1 is very naïve from a research perspective.
Response 6: This section has been repeated to adapt it to the research context. We set aside the engineering aspects. Figure 1 has been used to focus on the modules at the level of the recommendation engine which is the core of our system (cf page 6).
Point 7: Use case diagrams and the database design details are related to the application, not the research. Which research problem we address in a paper is important.
Response 7: We tackle problems related to yield optimisation, lack of information to farmers in malian agriculture using recommender system. They are mentioned in the paper. We removed use case diagrams in the article.
Point 8: The scientific novelty of the paper is missing. The Cosine similarity and Choquet’s integral are not included as a contribution to this paper. So no need to over-explain them.
Response 8: We do not claim to develop new approaches for the similarity calculus. We used some that already existed, taking into account our needs. The novelty of our work lies in the fact that no research work has yet addressed the problem we have addressed. This also explains the difficulty of comparing our system to another in the same on objective criteria.
Point 9: In a research paper, we include the algorithms, not the codes: Example of the wrong usage: Algorithm for integral of Choquet.
Response 9: This point is fixed. We included the right pseudo-code of integral of Choquet in the article.
Point 10: A single citation is adequate for Cosine similarity, therefore section 3.2.3 is not required to be in the paper.
Response 10: This point is fixed in the article.
Point 11: In a research paper, we do not provide a demo of the application. Figure 6. Test of adding crops.
Response 11: This point is fixed in the article.
Point 12: You need to add a problem statement section to the paper which also includes the formal mathematical definitions.
Response 12: the problem statement is done in introduction section.
Point 13: Figure 7 must include information about the scientific modules, not schematic components.
Response 13: Figure 7 has been modified.
Point 14: The evaluation section must include the evaluation metrics, the evaluation method, and the baselines.
Response 14: These comments are taken into account in the new version. In particular, we compared our system to agronomists’ advices with a given input data. The main metric used is the scientific accuracy determined by mention. The majority judgement technique is well known as evaluation technique.
Point 15: In general, it is not clear what novelty has been applied in this paper to promote recommendation systems from a scientific perspective.
Response 15: The article featured new work of its kind, a recommendation system in agriculture. The results obtained were compared with those proposed by certain specialists in the field and researchers made the comparison. After analysis and interpretation, the recommendations offered by our system turned out to be better than those made by engineers. We presented this in the evaluation section (cf 4.1, 4.4.2).
Point 16: The authors should do an experiment to compare the performance of the proposed method with another state of the art or at least common methods.
Response 16: We sought others solutions in the state of the art. But they don’t address the same problem like us. To the best of our knowledge, our system is the first to solve yield optimisation problem using some specificities related to agriculture in Mali. Having no developed systems doing the same thing, we considered two systems from two groups of agronomists to make recommendations that we compared to those produced by our system (see section 4.2.1).
Point 17: Appendix of an article usually includes the mathematical proofs.
Response 17: We do not have any mathematical proof to do in this case because our study is experimental.
Point 18: The scientific contribution of this paper is very limited.
Response 18: In the new version, we highlighted our contribution. It’s mainly based on SyrAgri a recommender system helping Malian farmers to increase their production (cf 3.1 and 4.2).

Round 2
Reviewer 1 Report
As the authors have followed the reviewers' suggestions, the article is publishable.
I do not have any more comments on the article.
Author Response
Responses to Reviewer 2 comments
Point 1: You need to revise your writing to make it more readable.
Response 1: We made effort in this context. We revised our writing by an english speaker.
Point 2: The significant novelties of the proposed system are not evident. What can be explained more about the contribution of the algorithm in the introduction section?
Response 2: We highlighted our contribution in introduction section. It’s about a recommender engine and a comparison between our system and agronomists' advices.
Point 3: Use case diagrams and the database design details are related to the application, not the research.
Response 3: We moved this part to the appendix A.
Point 4: I still advise that you add a clear mathematical problem statement with relevant notions.
Response 4: Our work is about a design of system to address concrete problems related to agriculture in Mali. Mathematical formulation of these problems is out of our knowledge.
Point 5: Define the problem in an independent section called problem statement, not the introduction. In the introduction, we explain the research concisely. So, in the problem statement, you can first define some definitions and then continue with the problem.
Response 5: This problem has been fixed. We added a section on problem statement.
Point 6: Add a section for the baselines, where you explain the baseline clearly. Also, add a section for the benchmark.
Response 6: Before our work, there was no system helping farmers to improve their yield. So, the unique baseline to which, we make comparison remains agronomists' advices. We already explained these advices in evaluation section.
Reviewer 2 Report
The authors have made several improvements to the paper. However, I advise them to consider the following points too.
Improve the writing of the paper:
Example below:
This research effort has the merit of having succeeded in involving ten (10) agricultural engineers who know the field and ten (10) agricultural researchers with a number of years of experience which varies between 2 and 35 years....
You need to revise your writing to make it more readable.
Moreover, from a research perspective, I do not advise you to have sentences like the above. Our method is good because engineers say that. Kind of irrelevant to the research. In research we explain the benchmark to justify the performance of the proposed method.
No need to have bullet points for the question in the introduction. Instead, add bullet points for contributions. Write three clear contributions to this paper in the introduction section.
The significant novelties of the proposed system are not evident. What can be explained more about the contribution of the algorithm in the introduction section? Add the bullet points.
Point 7: Use case diagrams and the database design details are related to the application, not the research. The research problem that you address in a paper makes the difference.
I still advise that you add a clear mathematical problem statement with relevant notions.
Authors mention in response number 9: The novelty of our work lies in the fact that no research work has yet addressed the problem we have addressed.
Define the problem in an independent section called problem statement, not the introduction. In the introduction, we explain the research concisely. So in the problem statement, you can first define some definitions and then continue with the problem.
Moreover, The class diagram is for DBMS. Not sure if it helps from a research perspective. I refer to Figure 2.
Add a section for the baselines, where you explain the baseline clearly.
Also, add a section for the benchmark.
Even though the authors claim that their study is experimental, I do not see any substantial experiments including, the impact of parameters or comparison with other contemporary competitors.
Author Response

(The authors gave the same response as above.)
